# FlexCap: Generating Rich, Localized, and Flexible Captions in Images

## Abstract

We introduce FlexCap, a module that generates localized descriptions for any region in a given image. We use the idea of length conditioning to ensure the output captions have the desired length. This allows for controllable generation of the full spectrum of localized captions, ranging from short object names to full sentence descriptions. To train this model, we create a dataset of image-box-caption triplets from web-scale text-image pairs using open-vocabulary object detection models. We show that FlexCap can connect images with LLMs by representing images as a sequence of region descriptions and their spatial extents. Using this interpretable textual representation, we exceed the state-of-the-art zero-shot performance on many visual question answering tasks. We also show that FlexCap can be fine-tuned to achieve strong performance on the dense captioning task on the Visual Genome dataset. Finally, we demonstrate qualitatively how FlexCap can be used for image labeling, object attribute recognition, and visual dialog.

## 1 Introduction

The groundbreaking success of large language models (LLMs) has led to the widespread adoption of this technology in a variety of applications. The introduction of vision-language models (VLMs) has further enhanced these capabilities by enabling reasoning over visual content in the form of question answering and visual dialog. Nevertheless, the ideal representation for utilizing LLMs to enable visual applications is still an open question.

A popular and effective strategy is to directly provide visual features as input tokens to LLMs. Two successful systems, Flamingo (Alayrac et al., 2022a) and BLIP (Li et al., 2022; 2023), have focused on utilizing latent representations produced by frozen visual backbones and integrating these with frozen LLMs. While Flamingo uses cross-attention to adapt visual tokens to be used by the language model, BLIP adapts visual tokens to be ingested just like text tokens by the language model. In this work we investigate if instead of representing the image by latent visual features, we can directly provide *textual representations* of an image by focusing on its elements: objects and regions.

We explore this alternative strategy with the idea of *flexible captioning* – generating controllably rich and localized captions as shown in Fig. 1. Our model, FlexCap, enables spatially controllable inquiry of any bounding box in the image, with the desired text detail controlled by the generated word count. FlexCap effectively combines three tasks that have been studied in isolation until now: image captioning, object detection, and dense captioning. While image captioning models can capture coarse semantic information they lack spatial understanding of the visual content. On the other hand, object detection brings spatial information in the form of bounding boxes. But object detection systems lack semantic details (attributes and relationships between objects) and are usually limited to a few classes. Recently open-vocabulary object detection increases the semantic diversity but it is still limited by text queries or prompts provided manually. Dense captioning (Johnson et al., 2016) is the task of localizing salient regions of the image and describing them with natural language sentences. However the capacity and richness of these models are limited with existing image captioning datasets (e.g. COCO(Lin et al., 2014) and Visual Genome (Krishna et al., 2017)). Moreover, these methods focus on full sentence generation rather than a mix of short and long captions at several levels of richness as people typically do. FlexCap combines all three tasks into one system by formulating each of them as different captioning problems – image captioning implies using FlexCap to caption the whole image as one big bounding box, dense captioning can be performed by conditioning on individual boxes, and object detection can be performed by prompting the model to produce short class names as captions.

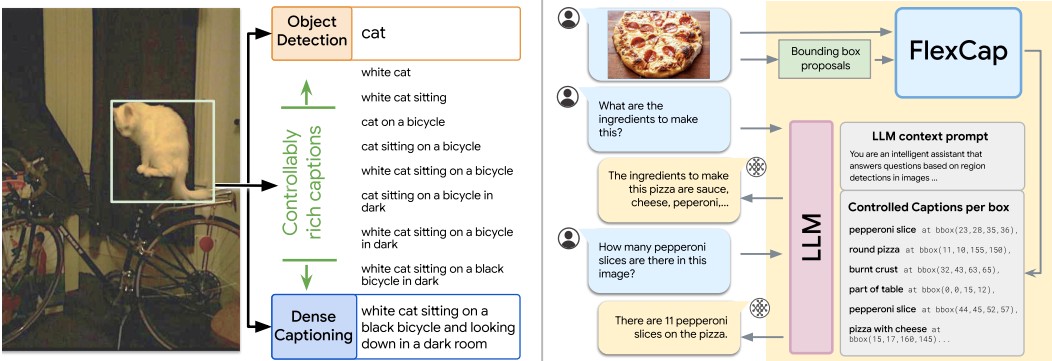

Figure 1: **FlexCap** generates controllably rich localized descriptions for any region in an image as shown on the left. It has the flexibility to produce captions in a controllable manner which allows the full spectrum of valid descriptions to be explored from short object category names to fully-detailed captions. On the right, we demonstrate that rich localized captions generated by FlexCap, when coupled with large language models (LLMs), enable zero-shot visual question answering.

To be able to train such a model, we require a dataset of images where many boxes are labeled with short and long descriptions. We propose a method to generate 32 billion triplets of (i) image, (ii) a proposed region within the image, and (iii) its corresponding caption from a web-based image-text pair dataset, through the use of open-vocabulary object detectors. Training FlexCap on this dataset enables the model to generate spatially and semantically rich representations (bounding boxes and their descriptions) that focuses on objects, their attributes, and their contextually changing descriptions. We show that this human interpretable representation, when combined with the power of LLMs, enables visual question answering and dialog. We also demonstrate that this combination can result in performance that is competitive with state-of-the-art VLM models on zero-shot image and video question answering benchmarks.

Our key technical contributions are: (i) controllable localized visual descriptions, using word count as a proxy for complexity to modulate the output of a generative language model, and using bounding box conditioning to indicate local regions in the image; (ii) a large-scale dataset generated from web-scale image-text pairs that enables training of our model; (iii) demonstrating that, with the support of LLMs, the human interpretable representation generated by FlexCap, is comparable performance of SOTA methods on open-ended image and exceeds SOTA performance on video question answering benchmarks in the zero-shot setup; (iv) demonstrating that our localized captioning performance exceeds the existing localized captioning methods under comparable scenarios.

## 2 FLEXCAP

**Architecture.** Our objective is to train a model that takes an image and a region of interest and outputs a description of the region spanned by the box. We present FlexCap's architecture in Figure 2. The model takes an image, the coordinates of a bounding box and the conditioning tokens as input, and outputs a textual description of visual contents within the specified bounding box. Our model mainly consists of an image encoder (i.e. ViT-B/16) and a transformer-based text-decoder. We pass the image through the vision model to produce outputs of dimensions $n \times d$ (where $n$ is the number of patches and $d$ is the embedding size ). We pass the bounding box coordinates (of dimensions $1 \times 4$) through a linear layer to produce the coordinate features (of dimension $1 \times d$). The vision features are concatenated with features from normalized bounding box coordinates. These concatenated inputs (of dimension $(n+1) \times d$) are fed into a text decoder which is a stack of $L$ Transformer layers. We use a decoder-only architecture in which all the vision, bounding box tokens remain unmasked but the text tokens are masked in a causal manner to enable next-word prediction training. The reason for adding all the vision tokens and bounding box coordinate tokens is so that the text decoder has access to all the context present in the image and the exact location of the bounding box. In this work, we train a decoder of 12 layers with a dimensionality of 768 and 12 attention heads. We use the standard vision transformer encoder layer architecture for the text decoder as well. In total, FlexCap has 248M parameters with 86M comprised of the image encoder (ViT-B) and the remaining parameters in the text decoder.

**Length conditioning.** For the same region there may be multiple valid captions. In the input image shown in Fig. 3, all the following descriptions are correct: *dog*, *Border collie*, *dog playing with a*

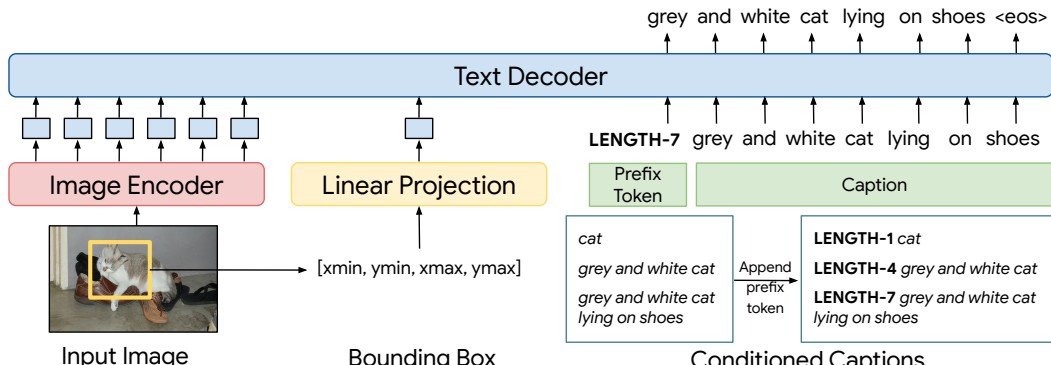

Figure 2: **Architecture and Training Setup.** We train a model that takes an image and a bounding box as input and outputs a length-controlled caption of the object contained in the bounding box. We specify the length by prefixing it before the caption. The training loss is the standard next-word prediction loss that is used to train image captioning models.

*frisbee*, *brown dog playing on a grass field*. This task does not have one right answer as for the same input there can be multiple correct outputs. We equip the model with the capability of producing outputs of a desired length by utilizing the idea of *length* conditioning. This is implemented simply by adding an additional token that indicates the desired length of the output caption. There are several advantages for doing this. First, the number of words used to describe is often proportional to the information content. We train the model to predict the next words in the sequence while accounting for the desired length, thereby the model learns to modulate the amount of information in the generated text. Also, length conditioning allows users to control the output of the model further enabling the use of a single model for many diverse tasks.

Additionally, the length prefix provides a better conditioned initial state for the captioner. Fig. 2 shows how the same box might have more than one ground truth captions ` a cat <e>` or ` grey and white cat lying on shoes <e>`. If we use the first caption as ground truth and the words ` a cat` as the seen text, the next-word prediction loss encourages the model to increase its score for the `<e>` token and decrease the score for the word `playing` due to the softmax loss. This is in direct conflict with the second caption. These kind of training issues are alleviated by the use of the length prefix.

**Loss.** We train the model to predict the next token of the text. The text tokens are prefixed with the desired length of the caption and appended with an end of sentence token `<e>` to indicate the end of the caption. The target text tokens are obtained by shifting the padded text by 1. This is common training methodology for training generative language models like GPT (Brown et al., 2020) or SimVLM (Wang et al., 2021). The loss is a classification loss over all the words present in the vocabulary. The loss is ignored over the padded tokens that are used to keep the size of the outputs same for all the captions in the batch.

**Implementation.** We implement this model using the JAX framework (Bradbury et al., 2018). We train the entire model from scratch for about 250K steps using the AdamW optimizer with a cosine learning rate schedule. The maximum learning rate is $6.4 \times 10^{-4}$ with 10K warm-up steps. We use a weight decay of $0.1$. We train with a batch size of $16384$ and image resolution of $224 \times 224$. We use a maximum text sequence length of $16$. For each image in the batch, we sample a maximum of 8 bounding boxes for each training step.

**Inference.** At inference time, we provide an image, the target bounding box, and the desired length as input. We then decode in an auto-regressive manner till the end of caption token `<e>` is encountered or the maximum number of decoding steps is reached. It is also possible to provide a partial caption as a prefix (such as *"this is made of"*) to the model which enables us to query for any particular property displayed in a given bounding box. We use standard sampling techniques used in text-generation like beam search, temperature sampling, or nucleus sampling (Holtzman et al., 2019) to generate multiple captions.

## 3 WEB-SCALE LOCALIZED CAPTIONING DATASET

In order to train the FlexCap model, we build a large scale dataset of image region descriptions of varying lengths. In the following section we describe how we produce such a dataset from existing

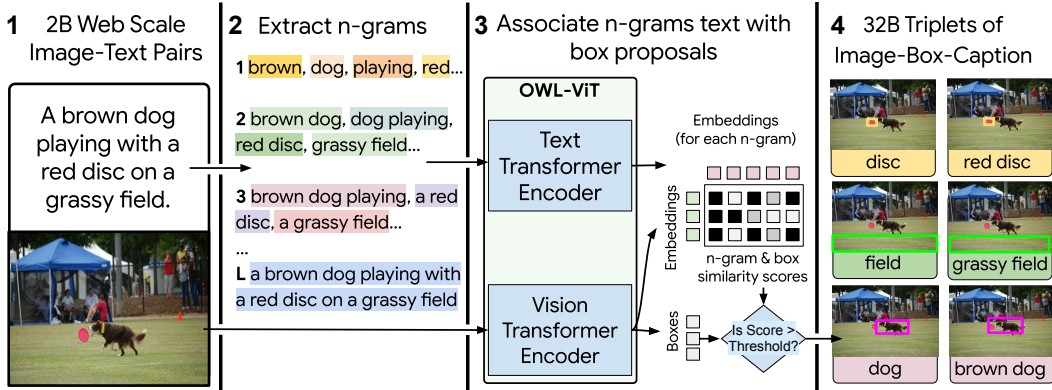

Figure 3: **Dataset Generation.** We use OWL-ViT to generate a dataset of triplets of image, bounding box and captions from a web-scale dataset of noisy image-text pairs. Increasing levels of richness in captions is captured through different length descriptions for each box.

image-text paired datasets. We leverage the web-based image-caption pairs from WebLI (Chen et al., 2022). The dataset generation pipeline is shown in Fig. 3. First we create *text queries* using n-grams from the caption of the image: e.g. "dog","brown dog", "brown dog playing with a disc". We specifically create n-grams where $n = \{1, 2, \cdots, 8\}$ and then filter out incomplete captions like "with a red", "dog playing with". More details about the filtering step are mentioned in the appendix. Then we use the filtered n-grams as *text queries* for pre-trained region proposal models (i.e. OWL-ViT (Minderer et al., 2022)) to extract boxes and select text-box pairs based on the similarity score ($> 0.1$). Multiple n-grams may match for a box, and this results in several ways of describing a box in the image as shown in Col. 4 in Fig. 3.

This data collection technique results in 32 billion image-box-caption triplets from 2 billion images without requiring extensive manual annotation. Our captions show a rich vocabulary that is close to common language used to describe objects in the context of an image. If we use MS-COCO's vocabulary then all humans in the dataset would get labeled as *person*. However by building our vocabulary in a bottom-up manner we end up with captions that contain more informative words such as *baby*, *nurse*, *policeman*, *firefighter*, or *baseball player* to describe the *person* class. Please refer to the appendix for details of dataset statistics and examples.

## 4 FLEXCAP WITH LLMS

Recent work on improving the recognition and reasoning capability of vision models, so that they can perform tasks such as question answering and visual dialog, has focused on connecting features from vision models like ViT to Large Language Models (LLMs). Flamingo (Alayrac et al., 2022b), PALI (Chen et al., 2022), PALM-E (Driess et al., 2023) have focused on using patch features from images as words or tokens. Here, we propose an alternate approach. Instead of adapting LLMs to understand vision features, we explore connecting images with LLMs using rich text in the form of *localized descriptions*. As FlexCap solves the rich visual perception problem by generating dense localized information, it enables LLMs to perform high-level spatial and textual reasoning by linking spatially enriched visual concepts with the world knowledge. When combined with their well-trained common-sense reasoning, LLMs can reason about attributes by looking at singular object detections, scene-level understanding by utilizing larger window descriptions, counts by checking number of bounding boxes with similar content, and relative spatial relationships by comparing location of the boxes. Now we will discuss how to connect FlexCap with LLMs.

**FlexCap-LLM.** To adapt the base FlexCap to have improved detection skills, output longer sentences, and identify OCR, we co-train FlexCap for 25k more steps on detection (COCO, VOC, OpenImages, LVIS), captioning (COCO Captions, Visual Genome) and OCR datasets (WebLI). For image captioning datasets we use the bounding box that covers the whole image. For decoding OCR, we use an OCR token. We find this co-training step useful for downstream tasks using the LLM. In Figure 4, we show how we use FlexCap with an LLM to solve visual questions. First, we convert an image to a sequence of localized descriptions that describe the image in terms of the objects and regions present in the image. To do so, we need region proposals. We use OWL-ViTv2 (Minderer et al., 2023) to localize important objects and regions in an image. We keep the top 64 bounding boxes by their *objectness* scores. We then use FlexCap to describe each box in the image in the

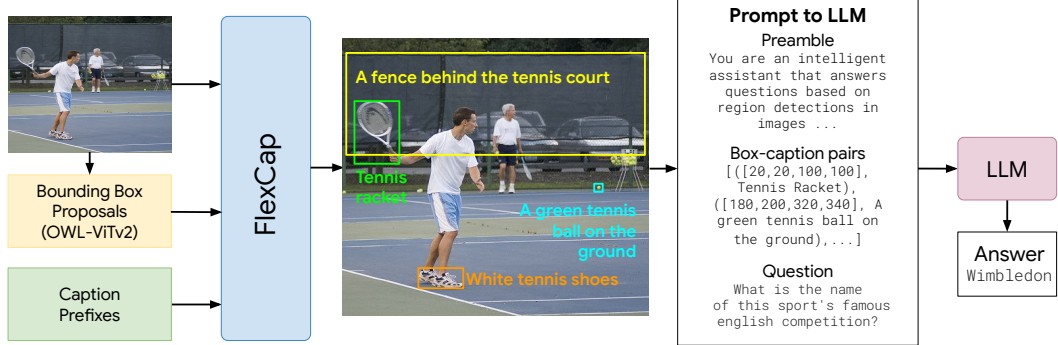

Figure 4: **FlexCap for VQA with bounding box proposals and an LLM**. FlexCap generates captions for different regions in a given image. To answer any open-ended questions, we prompt an LLM with FlexCap's detections (box-caption pairs).

context of the entire image. In order to produce holistic descriptions, we use multiple prefixes for each region. These prefixes are a combination of length conditioning token and some initial text. We add the boxes and their descriptions to a *text preamble* (see Fig. 4) that defines the setup where we are using an LLM to answer questions about an image. In all the experiments, we use PALM2-S model (Anil et al., 2023) as the LLM of choice. We refer to this end-to-end system that takes an image and a question to output the answer as *FlexCap-LLM*.

## 5 EXPERIMENTS

We evaluate our model across different tasks that require different levels of detail. In Section 5.1, we test FlexCap-LLM on visual question answering tasks. In Section 5.2, we evaluate how FlexCap can be used for localized captioning In Section 5.3, we provide ablations, qualitative results and applications of FlexCap.

### 5.1 VISUAL QUESTION ANSWERING

Visual question answering (VQA) often requires visually grounded rich semantic understanding of the content at multiple levels of granularity depending on the question. These properties make VQA a great test-bed for our method which can generate dense spatially grounded information on visual content with desired semantic complexity. Hence we evaluate the effectiveness of FlexCap-LLM on several image VQA benchmarks such as OKVQA (Marino et al., 2019), GQA (Hudson & Manning, 2019b), and VizWiz (Gurari et al., 2018), and video question answering benchmarks such as MSRVTT (Xu et al., 2016) and MSVD (Xu et al., 2017). Diverse characterictics of these datasets helps gaining better insight on FlexCap's capabilities. We report the commonly used accuracy metric for each dataset.

#### 5.1.1 IMAGE QUESTION ANSWERING

First we evaluate FlexCap-LLM on GQA, OKVQA and VizWiz image VQA benchmarks in a zero-shot setting, meaning that our approach is not trained with the task or the corresponding dataset. The results on these benchmarks are presented in Table 1.

**Compositional VQA**. GQA dataset is generated for evaluating the performance on complex compositional questions. As FlexCap produces information for multiple visual elements in the scene with their corresponding locations, it is quite well-suited for questions on compositional understanding of the image. On this benchmark, as shown in Table 1(a), FlexCap-LLM outperforms all the recent baselines except for ViperGPT (Sur'is et al., 2023) which is also known for its compositional understanding properties. Though, note that ViperGPT uses multiple tools on the fly with certain associated costs whereas we only utilize FlexCap outputs and feed them to an LLM.

**VQA with External Knowledge**. OKVQA dataset is particularly designed for evaluating the ability to answer questions about images that require external knowledge which is not readily available on the image. Hence it requires multiple levels of understanding of the content and reasoning with that informationm which is well-suited for applying FlexCap. Its performance on OKVQA is comparable to Flamingo and ViperGPT which highlights the effectiveness of the mix of generic and

Table 1: **Zero-shot image question answering results**. FlexCap-LLM is compared against recent baselines. Grayed out methods are trained on question answering datasets.

(a) **GQA results** on test-dev set

| Method | Accuracy(%) |
|---|---|
| LGCN (Hu et al., 2019) | 55.8 |
| LXMERT (Tan & Bansal, 2019) | 60.0 |
| NSM (Hudson & Manning, 2019a) | 63.0 |
| CFR (Nguyen et al., 2022) | 72.1 |
| FewVLM (Jin et al., 2021) | 29.3 |
| BLIPv2 (Li et al., 2023) | 44.7 |
| ViperGPT (Sur'is et al., 2023) | **48.1** |
| FlexCap-LLM | 47.5 |

(b) **OKVQA results** on Val set

| Method | Accuracy(%) ↑ |
|---|---|
| PalmE-12B (Driess et al., 2023) | 55.5 |
| PalmE-562B (Driess et al., 2023) | 66.1 |
| PaLI-3B (Chen et al., 2022) | 52.4 |
| PaLI-17B (Chen et al., 2022) | 64.5 |
| BLIPv2 (Li et al., 2023) | 45.9 |
| Flamingo (Alayrac et al., 2022b) | 50.6 |
| ViperGPT (Sur'is et al., 2023) | **51.9** |
| FlexCap-LLM | 50.6 |

(c) **VizWiz results** on Val set

| Method | Accuracy(%) ↑ |
|---|---|
| Flamingo 32-shot (Driess et al., 2023) | 49.8 |
| Flamingo FT (Driess et al., 2023) | 65.7 |
| PaLI-3B (Chen et al., 2022) | 67.5 |
| PaLI-17B (Chen et al., 2022) | 74.4 |
| Flamingo (Alayrac et al., 2022b) | 31.6 |
| FlexCap-LLM | **38.0** |

Table 2: **Zero-shot video question answering results** reported on MSRVTT-QA and MSVD-QA on test set. FlexCap-LLM is better than other zero-shot baselines for video VQA benchmarks.

| | MSRVTT-QA | MSVD-QA |
|---|---|---|
| Flamingo 3B (Alayrac et al., 2022b) | 11.0 | 27.5 |
| Flamingo 9B (Alayrac et al., 2022b) | 13.7 | 30.2 |
| Flamingo 80B (Alayrac et al., 2022b) | 17.4 | 35.6 |
| FlexCap-LLM (8 frames) | **24.9** | **40.2** |

specific descriptions generated by FlexCap. Unlike other baselines which use the question, FlexCap generates the captions without having access to the question .

**VQA with atypical images**. We also evaluate on VizWiz, which contains visual questions asked by people who are visually impaired. Unlike web content, in these images the objects and the scene are not always well-centered hence this dataset contains many out-of-distribution samples compared to typical web-crawled datasets. Nevertheless, our approach significantly outperforms Flamingo (Alayrac et al., 2022a) which also reports zero-shot performance on this dataset.

### 5.1.2 VIDEO QUESTION ANSWERING

We also evaluate FlexCap-LLM on zero-shot video question answering datasets MSRVTT-QA and MSVD-QA (Xu et al., 2017). The results on these benchmarks are presented in Table 2. For processing the video, we sample 8 frames uniformly from the video. We pass each of these frames through FlexCap to produce captions of objects and regions. We then combine all the object captions from the different frames into one prompt for the LLM. We observe FlexCap-LLM significantly exceeds the performance of the Flamingo 80B model in the zero-shot setting.These results highlight the zero-shot effectiveness of our method, which can solve tasks in the video domain even though both FlexCap and the LLM were not trained for those tasks.

### 5.2 DENSE CAPTIONING

**Dataset and Evaluation Metrics.** The dense captioning task is defined as producing both the regions and the corresponding descriptions for each region. For this experiment, we use the Visual Genome (Krishna et al., 2017)) dataset. In this dataset, each image is annotated with multiple

Table 3: **Captioning boxes in Visual Genome dataset.** FlexCap exceeds performance of other methods. All methods have been fine-tuned on Visual Genome captions.

| Method | mAP |
|---|---|
| FCLN (Johnson et al., 2016) | 27.11 |
| CAG-Net (Yin et al., 2019) | 36.29 |
| FlexCap | **43.62** |

(a) Captioning GT Boxes

| Method | mAP |
|---|---|
| FCLN (Johnson et al., 2016) | 5.39 |
| JIVC (Yang et al., 2017) | 9.31 |
| COCG (Li et al., 2019) | 9.82 |
| CAG-Net (Yin et al., 2019) | 10.51 |
| TDC+ROCSU (Shao et al., 2022) | 11.49 |
| GRiT (Wu et al., 2022) | 15.52 |
| FlexCap + GRiT Boxes | **15.61** |

(b) Dense Captioning

Table 4: **Compliance metrics.** FlexCap produces length-compliant captions for different lengths.

| Desired Length | Mean of Pred. Length | Std. Dev. of Pred. Length | Accuracy | Desired Length | Mean of Pred. Length | Std. Dev. of Pred. Length | Accuracy |
|---|---|---|---|---|---|---|---|
| 1 | 1.00 | 0.00 | 1.00 | 5 | 5.04 | 0.21 | 0.98 |
| 2 | 2.00 | 0.00 | 1.00 | 6 | 6.05 | 0.21 | 0.95 |
| 3 | 3.07 | 0.25 | 0.93 | 7 | 7.02 | 0.20 | 0.96 |
| 4 | 4.02 | 0.13 | 0.98 | 8 | 8.01 | 0.18 | 0.98 |

bounding boxes and each box has a corresponding caption. We use the train-test splits and evaluation metric as proposed in (Johnson et al., 2016). The paper proposes to use a mean of Average Precisions (mAP) over pairwise thresholds of both IOU thresholds (0.3, 0.4, 0.5, 0.6, 0.7) and Meteor score thresholds (0.0, 0.05, 0.1, 0.15, 0.2, 0.25). We use the same preprocessing of text and boxes as mentioned in (Wu et al., 2022).

**Fine-tuning FlexCap.** We fine-tune the pretrained FlexCap model on the Visual Genome train split for 60k steps with a lower learning rate of $1e-6$ at a resolution of $448 \times 448$.

**Captioning GT boxes**. Following the evaluation procedure from (Johnson et al., 2016), we evaluate captioning of the ground-truth boxes in Visual Genome. Since this setting removes the localization task, we have a cleaner evaluation of only the region captioning problem. The results of this experiment are provided in Table 3a in which we show that FlexCap achieves better performance compared to other approaches evaluated in this setting.

**Captioning GRIT boxes**. In this experiment, we want to compare against other approaches that perform both localization and captioning. We are measuring how well FlexCap performs when deployed together with an object detector. Table 3b shows FlexCap obtains better performance compared to other approaches evaluated in this setting. Since, in our work we do not propose any localization module we use GRIT's (Wu et al., 2022) region proposals as the input boxes for our model. This also allows us to directly compare our captioning capabilities against GRIT. We find that our approach outperforms GRIT at this task even though we test at a lower resolution of $448 \times 448$.

## 5.3 ABLATIONS AND QUALITATIVE RESULTS

**Compliance Metrics.** In this experiment, we measure how well our model complies to the desired caption length. To do so, we take 1000 images from MS-COCO dataset and use a random object in the image to produce a description with different lengths. We report the average length of the predicted caption, standard deviation, and fraction of times the predicted caption has a length equal to the desired length in Table 4. We observe that our model produces captions in a controllable manner with minor deviations. In Figure 5, we show qualitative examples of the FlexCap model producing different length captions for the same box. Note how the model progressively adds more information about the object by incorporating context in the longer sentences (*in the jungle*), attributes (*pink flamingo kite*), and alternative nouns (*chevy*, *feline*).

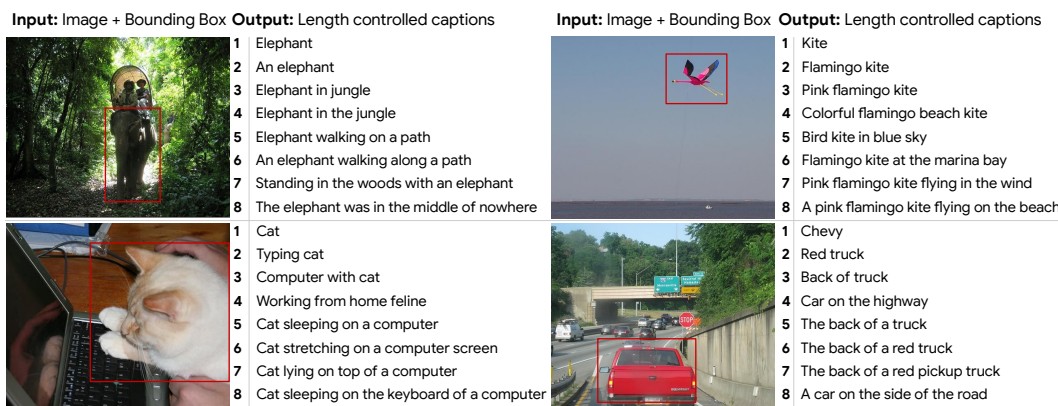

Figure 5: **Examples of length controlled captions generated by FlexCap.** Note that attributes ("pink flamingo kite") and context ("in the jungle") are generated as the length increases.

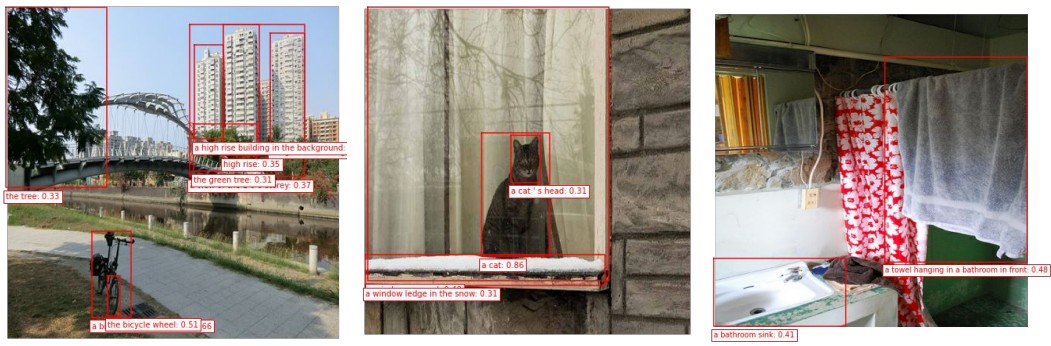

Figure 6: **Zero-shot Dense captioning results.** We show detections and generated captions on the Visual Genome dataset.

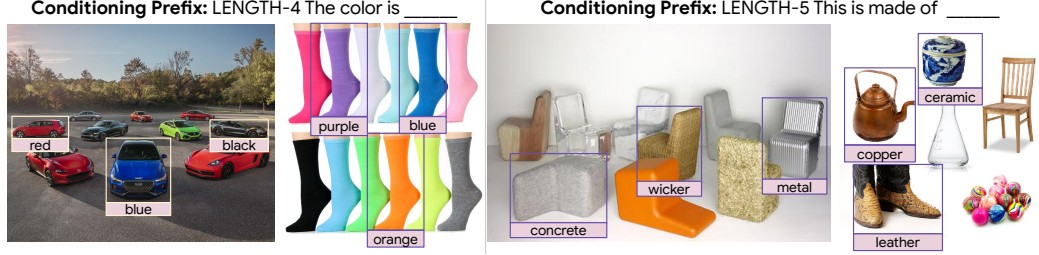

Figure 7: **Extracting properties by conditioning FlexCap with prefixes.** Examples of FlexCap extracting properties of objects of different categories by using relevant prefixes. Note how we are able to retrieve a one-word answer from the model by controlling the length of the caption.

**Zero-shot Dense Captioning.** We use OWL-ViTv2 to propose bounding boxes in the image. Flex-Cap then labels each bounding box with a caption. We show some qualitative results in Fig 6. We only show detections with a score $> 0.3$ to avoid clutter. These examples show how FlexCap can be used to detect and describe objects without the need for further training on this joint task.

**Extracting object properties.** Not only does FlexCap allow conditioning on the box and the length of the answer, it also allows textual prompting of a caption. We can use this property to our advantage to extract object properties such as color and material. We show examples of this in Fig. 7.

## 6 RELATED WORK

**Visual question answering** (VQA), a task designed to assess if a computer can answer questions about an image, often requires grounding visual concepts and reasoning. Although initially introduced for supervised evaluation of the task (Antol et al., 2015), most recently VQA also become one of the most powerful benchmarks for evaluating task and dataset independent visual dialog.

Several existing models such as ViperGPT (Sur'is et al., 2023), Flamingo (Alayrac et al., 2022a), BLIP (Li et al., 2022; 2023), PaLI (Chen et al., 2023) show convincing zero-shot performance on the VQA benchmarks that rivals the supervised approaches. Unlike most previous zero-shot approaches, which tightly couple vision and language components in a single model, FlexCap generates a high-level human interpretable representation of an image and demonstrates that, through straight-forward application of LLMs, we can achieve comparable performance with state-of-the-art results across VQA benchmarks. Unlike others, ViperGPT (Sur'is et al., 2023), also decouples vision and language components and reinterprets visual questions with LLM generated programs, and executes them using existing visual perception tools. Whereas, in our case we use only one powerful vision tool, i.e. FlexCap, to generate all the necessary information and leave the reasoning to an LLM. In that sense, FlexCap is quite complementary to ViperGPT as it can be one of the powerful tools that can improve the controllable visual understanding of the image for ViperGPT.

**Open vocabulary object detection** models like OWL-ViT (Minderer et al., 2022) and ViLD (Gu et al., 2021) enable the user to query any given text on the image and obtain matched bounding boxes for those queries. In these models the text is often encoded by a text encoder like CLIP (Radford et al., 2021) and T5 (Raffel et al., 2020). The text embeddings are compared with the category-agnostic box proposals coming from the visual backbone. In this work, we use OWL-ViT's text and vision encoders to associate bounding boxes with text-queries to produce our training data. By training a localized captioning model, we remove the manual step of providing per-dataset or per-image text queries to use OWL-ViT. RegionCLIP (Zhong et al., 2022) obtained good performance on open-vocabulary object detection by utilizing region-level vision-language contrastive learning on large scale data. We differ from this work as we generate the description for each bounding box instead of associating text queries (defined manually) with bounding boxes.

**Dense captioning** involves localizing salient regions of the image and describing them with natural language sentences, introduced in (Johnson et al., 2016). In practice, the existing work often produces longer and more informative descriptions of objects or their compositions using visual attributes of objects (Yin et al., 2019; Kim et al., 2019) or contextual and global image cues (Yang et al., 2017; Li et al., 2019). However, the richness of descriptions in this line of work are often limited to existing image captioning datasets (Lin et al., 2014; Krishna et al., 2017). By utilizing a large scale dataset of billions of noisy image-text pairs collected from the web (similar to (Jia et al., 2021; Chen et al., 2022)), we aim to generate more diverse sentences with a focus on describing the visual content in controllable detail using a richer visual descriptive space learned from the web.

**Length-controlled image captioning** has been explored in ZeroCap (Tewel et al., 2022) and LIC (Deng et al., 2020). ZeroCap (Tewel et al., 2022) implements length control as a post-processing step by changing the probability of sampling the end-of-sentence token. Hence the model is not naturally trained with word length conditioning in mind and cannot guarantee fine-grained length control at the level of number of words. On the other hand, LIC (Deng et al., 2020) generates length-controllable captions by conditioning the model with learned tokens that represent different length intervals. However there are considerable differences compared to FlexCap. First, our approach allows for controllability at the level of image regions, while LIC only provides full image captions. This is a significant difference, as it allows us to generate concise or detailed captions for all the objects in the image. Second, our approach has a more precise level of caption-length control. LIC uses a coarse subjective level of control with four or five levels of length (e.g. short, medium, long, and longer), while our approach allows for an exact number of words to be specified.

# 7 CONCLUSION

In this work we introduce FlexCap, a flexible captioning model that can describe localized regions in an image with controllably rich captions. To train FlexCap we generate a large-scale image-box-caption dataset that is rich in diversity of visual descriptions and their length. We achieve it by utilizing existing web-scale noisy image-text pairs and open-vocabulary object detection models. We show how localized rich descriptions provided by FlexCap can help us connect images and videos to LLMs and achieve strong performance on visual question answering and dense captioning tasks.

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

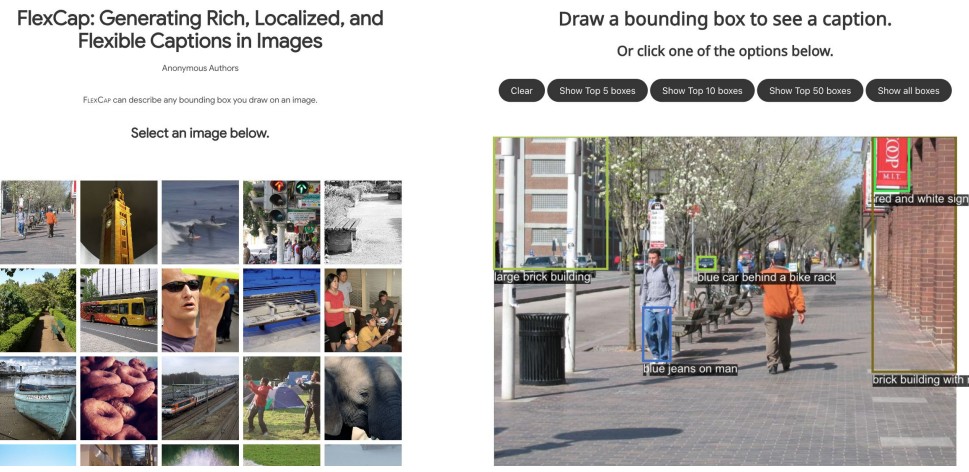

Figure 8: **Interactive demonstration of FlexCap.** Please visit https://flex-cap.github.io.

## APPENDIX

## A    INTERACTIVE DEMONSTRATION OF FLEXCAP

Through the attached webpage (please see `interactive_flexcap.html`), we demonstrate interactively how well flexible captioning works qualitatively on 40 images randomly selected from the Visual Genome dataset. The interface is shown in Fig. 8. Once the user clicks an image on the webpage, it shows the top-5 bounding boxes by default, but the user can draw any box to get a description for it using FlexCap. Since we cannot pre-compute all potential bounding boxes, we utilised OWL-ViT's box proposal network to identify 200 boxes and captioned them with FlexCap. The interface allows a user to draw a bounding box and then the closest pre-computed box is displayed if the IOU between the two boxes is above a certain threshold. We hope this interactive demonstration shows the capabilities of the FlexCap model in detecting and describing different objects and regions in an image.

## B    IMAGE-BOX-CAPTION DATASET DETAILS

Our dataset is composed of ~32 billion Image-Box-Caption triplets. In Figure 11, we show the distribution of caption lengths in the generated dataset. We observe that the distribution is not uniform. This is due to the fact that there are more n-grams of length 1 to sample than length 8. The average number of unique boxes in an image is 4.19, and the average number of captions per box is 4.04. We show some samples from the dataset in Figure 9. The alt-text from which the box captions are generated is provided as the title of the image. Note the alt-text gets clipped due to display-length limits which is why the detected boxes might have captions not visible in the displayed alt-text directly. We next discuss how captions of varying lengths are matched with different objects in an image.

**n-gram Filtering.** Before matching n-grams with boxes, we filter out n-grams that do not form informative or grammatically correct captions for boxes. This is done with three steps: 1) Removing any captions composed only of uninformative words (image, jpg, background, wallpaper, hd wallpaper etc.) 2) Removing n-grams that begin with words with which sentences usually do not start (of, on, in etc.) 3) Removing n-grams that finish with words with which sentences usually do not end (a, the, to, on etc.). This step is essential to reduce noise present in the large-scale image-text pair dataset obtained from the web. It is also important for the captioning model to produce grammatically correct informative sentences.

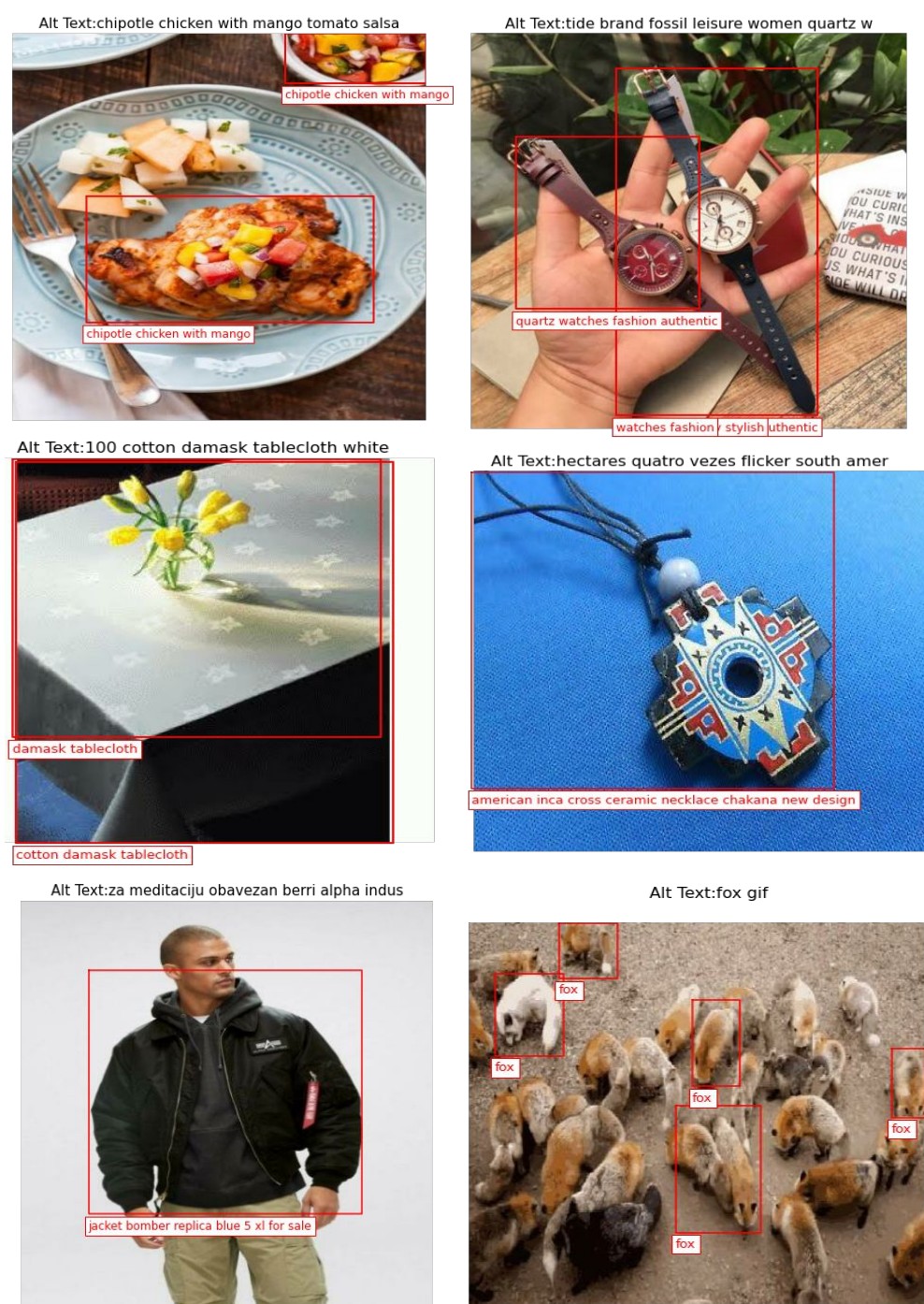

Figure 9: **Samples from the WebLI dataset Chen et al. (2022)** that are used for generating our Image-Box-Caption dataset. We only visualize a maximum of 5 boxes for each image to avoid clutter.

## C  OBJECT CLASSIFICATION

**Dataset and Evaluation Metrics.** In this experiment, we solely evaluate the recognition capabilities of our model in a zero-shot manner. We evaluate how good our model is at recognizing objects at

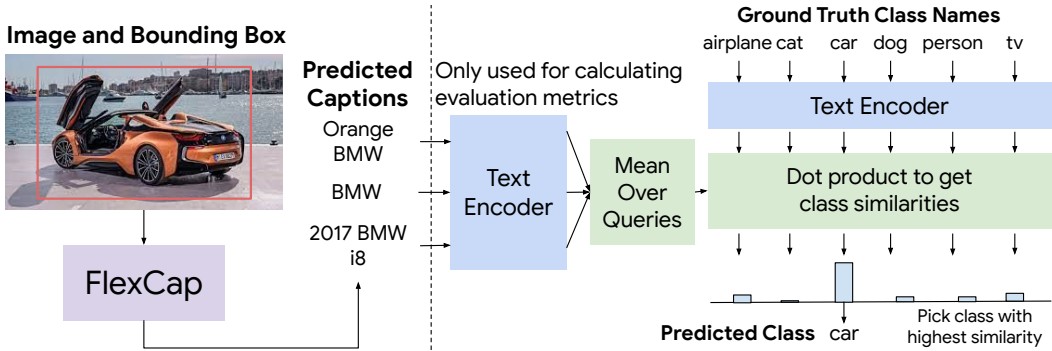

Figure 10: **Evaluating open-vocabulary outputs** from FlexCap with the help of the CLIP (Radford et al., 2021) text encoder.

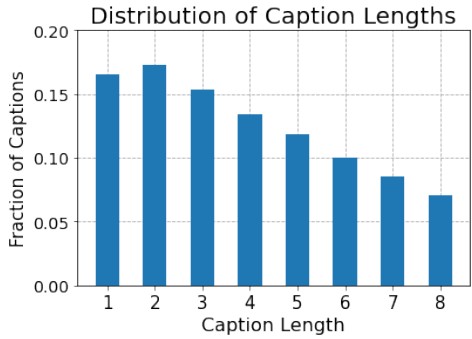

Figure 11: **Distribution of caption lengths in the Image-Box-Caption Dataset.**

different scales and under occlusion in object detection datasets. To do so, we consider the problem where an input image and the ground truth bounding box is provided as input to our model such that it produces a short description (1-4 words) of what is contained in the bounding box. We use 2 object detection datasets: PASCAL-VOC (Everingham et al., 2012) and MS-COCO (Lin et al., 2014) for this experiment. For each of these datasets, we present 2 metrics: the classification accuracy and the mean Average Precision (mAP) over all ground truth classes.

**Mapping Predicted Captions to Classnames.** Since our model produces captions for each box, we need to map the predicted descriptive sentences to classnames. To do so, we use an off-the-shelf text encoder (CLIP's (Radford et al., 2021)) to match the generated captions with known object class names. We show the full pipeline in Figure 10. We use nucleus sampling (Holtzman et al., 2019) to produce multiple outputs. We then take the multiple predicted captions and the ground truth class-names and pass both of these lists through the same text encoder. We then match the text embeddings of the predictions and the class names using cosine similarity of the query embedding and the class name embeddings. We report the results of this experiment in Table 5. Interestingly, the more descriptions our model generates the more accurate the predictions become. We show mAP improvements of 11% and 15.3% by just generating more caption samples per bounding box. CLIP (Radford et al., 2021) introduced the idea of using multiple prompts to get a more robust score for the class names, we show that generating multiple captions of the same image and averaging them provides a more accurate representation of the object as opposed to just one caption.

# D REFCOCO EXAMPLES

RefCOCO dataset Kazemzadeh et al. (2014) introduces a task which given an input text referring to an instance of an object in an image, the output is to localize the instance the text is referring to. It involves differentiating between instances of the same object category. On the other hand our model is trained for the inverse task of describing a given bounding box. However, we find images in this dataset to be good candidates to check how FlexCap describes same object classes using different words, particularly focusing more on attributes and context rather than 2D image positions (e.g. left, right) as is often done in RefCOCO annotations. Especially for differentiating people FlexCap usually relies on clothing and colors. We show qualitative results in Figure 12. We

Table 5: Classification accuracy and mAP increase considerably with number of captions generated by the model.

| No. of Captions | VOC | | COCO | |
|---|---|---|---|---|
| | Acc. | mAP | Acc. | mAP |
| 1 | 77.1 | 72.5 | 49.3 | 36.1 |
| 2 | 77.2 | 73.2 | 49.7 | 37.0 |
| 5 | 78.5 | 77.7 | 51.6 | 41.8 |
| 10 | 79.1 | 81.0 | 54.3 | 48.1 |
| 15 | 79.7 | 82.1 | 54.8 | 49.8 |
| 20 | 80.0 | 83.4 | 55.2 | 51.4 |

show ground-truths (green boxes) from RefCOCO to highlight what instances were chosen. In the first three rows, we show examples where FlexCap successfully describes different instances of the same category. In the last row we show two examples where FlexCap does not differentiate between different classes.

## E  VISUAL DIALOG

FlexCap-LLM can be used for the task of visual dialog Das et al. (2017). We first caption all the objects in the image using FlexCap. Once the image has been represented as the list of objects, we can interact with an LLM by providing the conversation turns as additional context for each query to the LLM. We show some examples of conversations with the FlexCap-LLM system in Figure 13. Note how the model is able to read text in the image in the leftmost figure, recognize material in the middle figure, and localize objects of interest in the rightmost figure. As we compute the object captions only once in the beginning of the conversations, there is no additional overhead of querying a large VLM for each additional turn in the conversation.

## F  LLM PROMPTS

We use the following prompts for the LLM in the question-answering experiments.

**OK-VQA and GQA.**

Preamble:`You are a helpful assistant answering questions about images to people.  You can look at the list of object detections in the image and answer questions.  The image content may not be sufficient to answer the questions, and you may need to rely on external knowledge resources or commonsense.  In an image, many objects were detected.  They are listed in the following format: [object descriptions] [cx, cy, w, h], where cx is x coordinate of the center, cy is the y coordinate of the center, w is the width and h is the height of the bounding box of that object in the image.  The list of objects is as follows:`

Object representation: `[captions] [cx, cy, w, h]`

Question Prompt: `Q: <question> Answer in one word.  A:`

**VizWiz**

Preamble:`You are a helpful assistant answering questions about images to people.  You can look at the list of object detections in the image and answer questions.  The image content may not be sufficient to answer the questions, and you may need to rely on external knowledge resources or commonsense.  In an image, many objects were detected.  They are listed in the following format:  [object descriptions] [cx, cy, w, h] [score], where cx is x coordinate of the center, cy is the y coordinate of the center,`

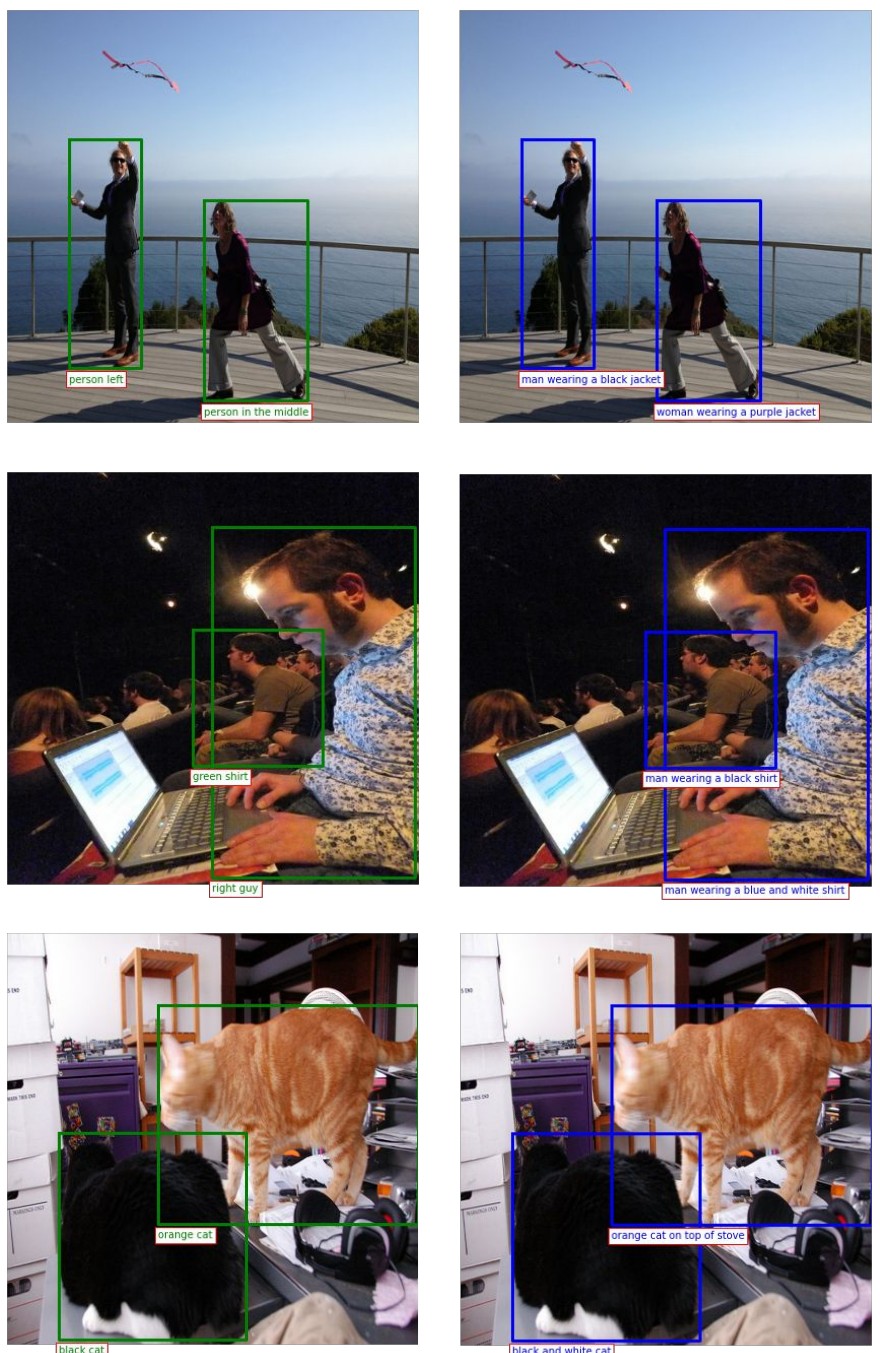

Figure 12: **FlexCap on RefCOCO Dataset.** For each image pair, RefCOCO ground-truth captions are displayed on the left, and FlexCap captions are displayed on the right. Our method focuses more on attributes and context rather than 2D image positions (e.g. left, right) as is often done in RefCOCO annotations.

```
w is the width, h is the height and s is the confidence score for
the object detection.  Low score means the detection is likely
inaccurate, and this often makes the question unanswerable.  You
```

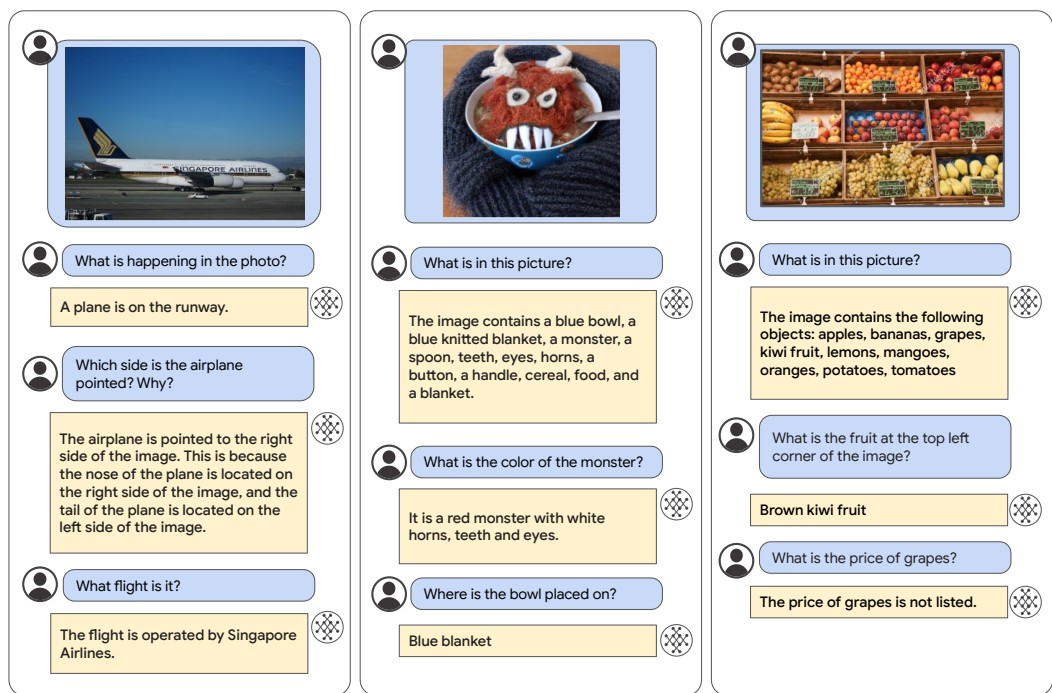

Figure 13: **FlexCap-LLM for Visual Dialog**

can answer questions as 'unanswerable'. The list of objects is as follows:

Object representation: `[captions] [cx, cy, w, h] [score]`

Question Prompt: `Q: <question> Answer in one word. A:`

**MSRVTT and MSVD.**

Preamble:`You are a helpful assistant answering questions about videos to people. You can look at the list of object detections in each frame and answer questions. In a video, many objects were detected in each frame. In frame <frame number>, the following objects were detected:`

Object representation: `[captions]`

Question Prompt: `Q: <question> Answer in one word. A:`

