# OpenReview forum: "FlexCap: Generating Rich, Localized, and  Flexible Captions in Images"
_ICLR.cc/2024/Conference — Submitted to ICLR 2024_

### Official Review · Reviewer_sAQH · 2023-10-30

**Soundness:** 4 excellent
**Presentation:** 4 excellent
**Contribution:** 3 good
**Rating:** 5
**Confidence:** 5

**Summary:**

This paper proposes a novel archtiecture to let LLM deeply understand an image in detail.

The proposed FlexCap consists of two important ideas:
(1) Train a captioning model that can generate captions for specific bounding box inside an image.
(2) Using Prompt engineering, this paper also propose a way to extend an LLM into a multimodal LLM by providing it different captions based on different bounding boxes.

By doing this, FlexCap-LLM is successfully turned into a multimodal agent that can perform various tasks including VQA.

**Strengths:**

The framework that combines LLM with a captioning model is a great alternative to save budget it takes to train a multimodal agent.

While previous works train a single model that can serve as a multimodal agent, authors utilize captioning model and LLM to build a multimodal agent.

While there can be possible performance degradation, this 2 stage model can still shows great performance compared to other one stage model.

**Weaknesses:**

While the 2-stage concept of FlexCap-LLM shows effectiveness in various downstream tasks, LLM can't directly understand an image in latent space.

From the result of this work and previous works, it's not still sure that 1 stage or 2 stage is better than the other.

FlexCap-LLM also relies on the performance of FlexCap's captioning performance. If FlexCap generates wrong caption for each bbox, this can lead to hallucination.

**Questions:**

As I mentioned in weakness section, FlexCap-LLM seems to rely on the captioning performance of FlexCap.

Is there any way to filter out mismatching caption with input bounding box?

---

> ### Author Response · Authors · 2023-11-20
> **Rebuttal response for Reviewer sAQH**
>
> We thank the reviewer for their effort and time reviewing our work. We address the concerns raised below:
>
> *W1. While the 2-stage concept of FlexCap-LLM shows effectiveness in various downstream tasks, LLM can't directly understand an image in latent space.*
>
> Yes, we agree that in this work the LLM does not directly understand the image in the latent space. But this might not be a limitation. First, we show that without having to train LLMs to understand image latents, we can get comparable or better zero-shot performance on visual question answering tasks by providing more informative input text. Hopefully this work inspires alternative research in ways to produce more informative inputs for LLMs. Whether it is image latents or text or a combination of both will remain to be seen.
>
> Furthermore, we believe a model like FlexCap will be useful in producing descriptive and localized image datasets that will be useful for training 1 stage models. Present image-text datasets are sparsely labeled in the sense that they do not describe all the objects and regions present in the image. We believe that work in this direction will lead to more densely labeled datasets.
>
> *W2. From the result of this work and previous works, it's not still sure that 1 stage or 2 stage is better than the other.*
>
> Perhaps rather than picking a winner on one metric, we can look at both approaches from multiple angles where one would be more advantageous than the other. For instance some advantages of 2 stage approach are:
> 1) Easy interpretability of the inputs to the LLM. If the object detection or localization is wrong we can work on better recognition or detection models to improve performance of the system.
> 2) Identify the source of hallucinations. Since it is a two stage process we can easily identify if the source of a hallucination is the vision or language model when it happens in a conversation with a user.
> 3) Using pre-trained LLMs for vision tasks shows we can improve performance for reasoning and question answering tasks by working on more informative inputs without the need for a large number of resources required to train language models together with visual inputs.
>
> *W3. FlexCap-LLM also relies on the performance of FlexCap's captioning performance. If FlexCap generates wrong caption for each bbox, this can lead to hallucination.
> ...
> As I mentioned in weakness section, FlexCap-LLM seems to rely on the captioning performance of FlexCap.*
>
> We agree with the reviewer that the final performance of FlexCap-LLM is dependent on both FlexCap and LLM’s performance. The novelty of our work is the FlexCap model and FlexCap-LLM is the name of the system used to evaluate the performance of the FlexCap model. FlexCap-LLM works by passing an image through FlexCap first to get many boxes and their respective captions. These localized captions are passed onto an already trained LLM. Through this experiment, we want to highlight the fact that the captions generated by FlexCap are good enough for solving downstream tasks with LLMs. While errors can occur during captioning, we find the combination of FlexCap+LLM is at par or better than other zero-shot baselines that are trained in an end-to-end manner (Table 1 and 2).
>
> The answers to the reviewer's questions are below:
>
> *Q1. If FlexCap generates wrong caption for each bbox, this can lead to hallucination.
> Is there any way to filter out mismatching caption with input bounding box?*
>
> Yes it is possible to use an open-vocabulary object detector like OWL-ViT to filter out any hallucinated captions. The text embedding of a wrong caption will have a lower matching score with the embedding of the object in the image. We use this method in the VizWiz experiments to indicate how well the predicted sentence matches with the region in the image. We did not find this score to be that useful for the other VQA tasks. We will add this detail to the paper.

---

### Official Review · Reviewer_N2tn · 2023-10-31

**Soundness:** 3 good
**Presentation:** 3 good
**Contribution:** 2 fair
**Rating:** 5
**Confidence:** 3

**Summary:**

This paper proposed a model to generate controllable localized visual descriptions of bounding box, and proposed a large-scale dataset generated from web- scale image-text pairs. The experiments show that FlexCap exceeds SOTA performance on several benchmarks in the zero-shot setup, and achieves superior localized captioning performance.

**Strengths:**

1. A large-scale dataset is generated by the author, which might enable the training of multi-modal tasks.
2. The experimental results are abundant, eg. visual question answering tasks, localized captioning, and the proposed model achieves promising results on several tasks.

**Weaknesses:**

1. The architectures are mostly a compilation of pre-defined visual models and large language models. However, the novelty and inspiration for the latter works are questionable.

2. Another concern is the precision or accuracy of the proposed large scale dataset. The authors claimed they use the filtered n-grams as text queries for pre-trained region proposal models (i.e. OWL- ViT (Minderer et al., 2022)) to extract boxes and select text-box pairs based on the similarity score (> 0.1). Therefore, the upper bound of precision of this dataset is determined by OWL- ViT, this might restrict the downstream tasks on the dataset.

3. The implementation details should be removed to the experiments section.

**Questions:**

1. The description about loss is imprecise, please use some formulations to formalized it. Additionally, the authors claimed that 'The loss is ignored over length-prefix tokens and...'. However, as shown in Figure2, the prediction of length-prefix tokens should be 'grey', the first token in the sentence. Is this ignored in the training process?
2. Why is the 'length conditioning' works? The length conditioning is implemented simply by adding an additional token that indicates the desired length of the output caption, and there is no training constrains. How to guarantee the additional token constrains the length?

---

> ### Author Response · Authors · 2023-11-20
> **Rebuttal response for Reviewer N2tn**
>
> We thank the reviewer for their effort and time reviewing our work. We address the concerns raised below:
>
> *W1: The architectures are mostly a compilation of pre-defined visual models and large language models. However, the novelty and inspiration for the latter works are questionable.*
>
> We use standard transformer-based architectures (which is quite common and has been successful in a variety of domains) for the vision encoder and the vision-bounding-box-text decoder. However, both the encoder and decoder are trained from scratch using our image-box-caption triplet dataset end-to-end for the localized captioning task. In other words our visual models may be considered pre-defined in terms of architecture, however, they are not pre-trained. The novelty in this work is that the resulting FlexCap model is a localized image captioner that can be conditioned using bounding boxes. We also propose a method to obtain the large-scale and diverse dataset required to train such a model.
>
> *W2: Another concern is the precision or accuracy of the proposed large scale dataset. The authors claimed they use the filtered n-grams as text queries for pre-trained region proposal models (i.e. OWL- ViT (Minderer et al., 2022)) to extract boxes and select text-box pairs based on the similarity score (> 0.1). Therefore, the upper bound of precision of this dataset is determined by OWL- ViT, this might restrict the downstream tasks on the dataset.*
>
> There is an inherent trade-off between precise, small-scale, well curated data (e.g. COCO captions) and large-scale diverse but noisy data (e.g. CLIP training set). The existing research on full image captioning (e.g. CLIP, CoCa, CAPPA) shows that large-scale noisy data could be more effective even if it is noisy data, mainly because it has diversity and enables training at scale. In this work we also choose to explore using a large-scale albeit noisy dataset as we want to generate a rich caption for any region in an image. While there will be a certain level of noise when using internet caption data, we find we can still train models with strong object localization using this data as our results demonstrate.
>
> The answers to the reviewer's questions are below:
>
> *Q1: The description about loss is imprecise, please use some formulations to formalized it. Additionally, the authors claimed that 'The loss is ignored over length-prefix tokens and...'. However, as shown in Figure2, the prediction of length-prefix tokens should be 'grey', the first token in the sentence. Is this ignored in the training process?*
>
> Thank you for catching this. We acknowledge that the loss is not ignored over prediction of length-prefix tokens (that is 'grey' in the example will be considered for the loss). We will remove the sentence “The loss is ignored over length-prefix tokens” from the paper .
>
> Below we formalize the loss function and will include in the paper:
>
> We have a triplet $T=(x, b, W)$ for image $x$, bounding box $b$ and box caption $W$.
>
> In our work,
>
> $$W={w_0,w_1,w_2,...w_M} = {Length_K,w_1,w_2,...w_M}$$
>
> where $Length_K$ is the number of words present in the box caption $W$. In the example in Figure 2, $W = [Length_7 $, grey, and, white, cat, lying, on, shoes, <eos>$]$.
>
> For a given data triplet $T$, our objective is to maximize the following log-likelihood.
> $l(T) = l(x, b, W) = \sum_{i=1}^{M} \log P(w_i| w_{<i}, x, b)$.
>
> Assume that we have a dataset $D = {T_1, T_2, ..., T_N}$. The overall loss function is:
> $L = \frac{\sum_{j=1}^N l(T_j)}{N} = \frac{\sum_{j=1}^N l(x_j , b_j, W_j)}{N}$.
>
> *Q2: Why is the 'length conditioning' works? The length conditioning is implemented simply by adding an additional token that indicates the desired length of the output caption, and there is no training constrains. How to guarantee the additional token constrains the length?*
>
> We train the entire model including the text decoder from scratch for the task of length conditioned caption generation. For dataset generation, we associate n-gram textual captions with bounding boxes (refer fig 3). During training, the length token is prefixed to each textual caption to indicate to the model the desired length of the caption. The training constraint of adhering to the length conditioning exists in the loss function. Throughout training the model needs to predict outputs of a certain length and then “<eos>” token (end-of-sentence) or it will result in a high loss.
>
> In Section 5.3, We measure how often the trained model complies to the length constraint. We observe that if we train with length conditioning the model complies with the desired length most of the time (97 out of 100). The detailed statistics on compliance can be viewed in Table 4. Our length-conditioning claims are mainly backed with strong empirical evaluations.

---

> > ### Comment · Reviewer_N2tn · 2023-11-22
> >
> > Thanks a lot for the response of authors. Some of my concerns have been solved. Although the experiments demonstrate the trained model could generate the sentences with proper length, however, it is still difficult to figure out why the length constraint works.
> > Thus,  I'd like to maintain my borderline rating.

---

> > > ### Author Response · Authors · 2023-11-22
> > > **Explanation of length-conditioning**
> > >
> > > We are grateful to the reviewer for their time and consideration. We would like to take another opportunity to clarify why the length constraint works.
> > >
> > > Let us start with our training data. In our dataset, for each box there exist multiple captions of varying lengths. In Fig 3, for the bounding box enclosing the dog there are 3 captions: dog (length 1), brown dog (length 2) and brown dog playing (length 3). To make it clear to the model that we expect captions of different lengths, we produce 3 ground truth sequences: [$Length_1$, dog,  <eos>], [$Length_2$, brown, dog, <eos>], [$Length_3$, brown, dog, playing <eos>]. The model is now trained to predict the caption given the image, bounding box, and the length conditioning prompt. If the model predicts a sentence that is longer than one word when conditioned with $Length_1$ token, the model incurs a high loss. The model will only incur a low loss if it learns to predict <eos> after a caption of length 1. If the caption is longer or shorter it incurs a high loss.
> > >
> > > Given the 32B triplets of image-box-caption as training data, the model sees a variety of captions of varying lengths. In Fig 11, we show the length distribution in the dataset which shows the diversity of length in our captions.
> > >
> > > After training exclusively on this objective, the model learns to predict captions by choosing the appropriate amount of  description for that particular length (as indicated by the length conditioning token). In Fig 5, we get a sense how the model is achieving this goal. Sometimes the model incorporates articles (elephant, *an* elephant), adjectives (cat, *typing* cat), verbs (computer with cat, cat *sleeping* on a computer), and more contextual information (bird kite in blue sky, colorful *flamingo* *beach* kite) to produce length-conditioned captions.
> > >
> > > We hope this explanation makes things clearer. We will be happy to explain any further questions regarding the length conditioning.

---

### Official Review · Reviewer_1skp · 2023-11-01

**Soundness:** 2 fair
**Presentation:** 3 good
**Contribution:** 2 fair
**Rating:** 5
**Confidence:** 4

**Summary:**

The work proposes a module called FlexCap, which can generate flexible-length of captions for a bounding-box. FlexCap is trained using a self-collected dataset containing variable length of captions. The experiments show that such captions can be used to prompt LLMs to achieve good performance for vision-language tasks.

**Strengths:**

- The work proposed a new end-to-end model which can generate a variable length (as condition) of caption of a region. Such a model is new (as far as I know), and can be beneficial for future research.
- The experiments demonstrates effectiveness of variable length captions by FlexCap.

**Weaknesses:**

-  In introduction, the authors claim that they diverge from Flamingo and BLIP2 (which use latent representations to connect LLMs), and use textual representation of images. However, the idea of using textual description of images to prompt LLMs for vision-language tasks is not new.
- The major contribution, comparing to prior works, is producing a desired length of caption. Thus a baseline is missing: Simply use a caption model to predict (1) whole image description, (2) a detection model to predict the object in the box, and prompt a LLM conditioned on (1) and (2) to generate desired length of captions. This baseline circumvents the collecting of a new dataset and training a model like FlexCap.
- In fact, I am wondering if other carefully filtered/designed localized captions could better prompt the LLM for those tasks. The work fails to compare or propose other baselines to demonstrate the superiority of FlexCap.

**Questions:**

- Will the dataset be released?
- Will the pre-trained model (weights) be released?

---

> ### Author Response · Authors · 2023-11-20
> **Rebuttal response for Reviewer 1skp - Part 1**
>
> We thank the reviewer for their effort and time reviewing our work. We address the concerns raised below:
>
> *W1: In introduction, the authors claim that they diverge from Flamingo and BLIP2 (which use latent representations to connect LLMs), and use textual representation of images. However, the idea of using textual description of images to prompt LLMs for vision-language tasks is not new.*
>
> We agree with the reviewer that “using textual description of images to prompt LLMs for vision-language tasks” is not new. What is new in our work is providing more informative textual inputs in the form of localized captions produced in a controllably-rich manner. To achieve this, we also had to create a large-scale dataset of localized captions that enables the training of such a model. We will clarify this in the paper.
>
> *W2: The major contribution, comparing to prior works, is producing a desired length of caption.*
>
> We would like to point out that besides producing a caption of desired length, a key contribution of our work is producing a localized caption that describes the content in any given bounding box using the context of the whole image. The captioning model needs to describe objects and regions of all sizes in the image.
>
> *W2: Thus a baseline is missing: Simply use a caption model to predict (1) whole image description, (2) a detection model to predict the object in the box, and prompt a LLM conditioned on (1) and (2) to generate desired length of captions. This baseline circumvents the collecting of a new dataset and training a model like FlexCap.*
>
> We thank the reviewer for this suggestion. In theory, this is a sound approach. However, we found that the current leading image captioning models (such as BLIP2) do not recover all the objects in an image. We experimented initially with this approach while trying to localize objects with captions generated by VLMs. However image captioning models usually only generate captions mentioning salient objects in an image. This is what inspired us to work on a model that generates descriptions for any region in an image instead of matching existing descriptions with regions.
>
> As an example we ran BLIP2 on the same images for which we have FlexCap results here:
>
> * [Image 1](https://flex-cap.github.io/images/00.html):
>
>     * BLIP2 captioner returns the following output: *two people walking down a sidewalk.*
>     * BLIP2 VQA with the prompt “List all the objects in the image.” returns the following output: *A man walking down the sidewalk, a man on a skateboard, a man on a bike, a man on a bench, a man on.*
>     * Objects that FlexCap described in the image (subset shown here): *brick-paved street, a black trash can, window, orange jacket, trees lining the street, blue jeans, a man is wearing a orange jacket, a bench on the sidewalk, a yellow car, red and white sign on the side of the building, a white pole on the sidewalk, a white street sign, a black bag on a man’s shoulder, car on the street.*
>
> * [Image 2](https://flex-cap.github.io/images/09.html):
>
>     * BLIP2 captioner returns the following output: *a group of people playing a video game.*
>     * BLIP2 VQA with “List all the objects in the image.” returns the following output: *people, a dog, a cat, a tv, a computer, a couch, a window, a door, a chair, a table.*
>     * Objects that FlexCap described in the image (subset shown here): *red shirt on a woman, brown couch, wii controller, watch, black glasses, cap, magazines, framed picture on the wall, white blinds on window, papers on table, a watch on man’s wrist, a pair of black shorts, remote strap, a stuffed animal on the couch, a stack of magazines, a framed picture on the wall.*
>
> As the first part of the suggested approach misses on detecting a large number of objects, an LLM will be unable to produce reasonable descriptions of varying lengths for all objects in the image.
>
> *W3: In fact, I am wondering if other carefully filtered/designed localized captions could better prompt the LLM for those tasks. The work fails to compare or propose other baselines to demonstrate the superiority of FlexCap.*
>
> We thank the reviewer for this feedback. We are also optimistic about using better localized captions to prompt LLMs. In Section 5, we explore how FlexCap’s outputs (localized descriptions) can be used to prompt existing LLMs for various VQA tasks.
> We also compare the usefulness of such localized captions for downstream tasks against other baselines such as Flamingo, BLIPv2 and ViperGPT. We think these are strong baselines that use LLMs in different ways for VQA tasks. We are happy to include comparisons with any particular baseline that the reviewer has in mind.

---

> > ### Comment · Reviewer_1skp · 2023-11-22
> >
> > Thanks for the responses. However, I am not fully impressed by the contributions of the work and maintain my borderline rating.

---

> > > ### Author Response · Authors · 2023-11-22
> > > **Rebuttal response for Reviewer 1skp**
> > >
> > > We appreciate the reviewer's suggestion of a good baseline that does not require collecting data or training new models. We have shown that our model (FlexCap)  is much more powerful than the suggested baseline (see more results [here](https://flex-cap.github.io/)). FlexCap not only describes more objects in an image than the baseline, but also describes object attributes, objects in the context of the image, and even the parts that make up the objects. Furthermore, we show that these descriptions are informative enough to be used to solve VQA tasks at par or better than baselines such as BLIPv2, Flamingo, and ViperGPT.
> > >
> > > We would be happy to present any additional evidence that the reviewer requires to address their concerns.

---

> ### Author Response · Authors · 2023-11-20
> **Rebuttal response for Reviewer 1skp - Part 2**
>
> The answers to the reviewer's questions are below:
>
> *Q1: Will the dataset be released?*
>
> We will not be able to release this dataset as it is based on WebLI, which is not yet open-sourced. However, we will release the code used to produce image-box-caption triplets from any image-text pair dataset (like YFCC100M). The data generation code is based on the OWL-ViT models, which are open-sourced. We believe the community will benefit from having a method to produce localized region-level captions from image-text pairs.
>
> *Q2: Will the pre-trained model (weights) be released?*
>
> We are actively looking into ways to release the model weights.

---

### Author Response · Authors · 2023-11-23
**General Response**

We are grateful to the reviewers for their time and consideration.

The reviewers identified the following strengths in the paper:

1. A new end-to-end architecture for variable length captioning of a region in an image (1skp).

2. Large-scale dataset which might enable training of multi-modal tasks (N2tn).

3. Combining LLM with a captioning model to train a multimodal agent (sAQH).

4. Extensive experiments demonstrating effectiveness of FlexCap (1skp, N2tn).

Based on the feedback of the reviewers, we added the following information during the rebuttal period:
1. Qualitative comparison with a baseline based on image captioning models that does not require additional training (asked by 1skp). We show how FlexCap is an improvement over the suggested baseline. Please see more examples [here](https://flex-cap.github.io).

2. Added more information about why the length conditioning works (asked by N2tn).

3. Added the formulation of loss function for FlexCap (asked by N2tn).

We have also responded to all the weaknesses and questions in detail below.

We are hopeful that the clarifications provided will enable the area chairs and reviewers to recognize the novelty of the FlexCap model and the impact of the applications it enables.

---

### Meta-Review · Area_Chair_suat · 2023-12-05

**Metareview:**

**Summary**

The authors propose a flexible-length image region captioning model. They also collect a dataset from web-scale image-text pairs, and a region based vision model (OWL-ViT) is used in the annotation process. This proposed region captioner can be further integrated with a text-only LLM to enable vision-language capabilities with carefully designed prompts. It achieves strong zero-shot performance on many benchmarks.


**Strengths**

- The variable length region captioning model is new.
- The dataset used to train it could also be a valuable resource.
- Abundant experiments are conducted.


**Weaknesses**

- model and dataset cannot be released
- lack of novelty -- use captioning to enable LLM for vision-language task is not new.
- the proposed dataset quality is limited by the pre-trained region model (OWL-ViT)
- some missing baselines (authors has attempted to address with some preliminary experiments in rebuttal)
- cascaded 2-stage model -- error in captioning may lead to LLM hallucination (initial solution via filtering with vision model is proposed in rebuttal)

**Justification For Why Not Higher Score:**

After the rebuttal and reviewers discussion, all 3 reviewers think this work does not bring novel ideas significant enough for ICLR publication, albeit the model and dataset proposed can be considered new from some reasonable perspective. (One reviewer was initially positive but later agreed with others and lowered the rating.) However, my major concern about this work is the lack of contribution to the community. It seems this work heavily relied on proprietary resources that prevent the community to reproduce / further build upon this work. The authors couldn't promise to release the data / model either. As an AC, this is my major deciding factor to reject this submission. After all, if only very limited people can build upon this work, the contribution is kind of limited to some benchmark numbers and textual description of method that is difficult to code up. ICLR has been known for its open-ness philosophy of research. Although full release is not a hard requirement, the lack of novelty and limited contribution still justify the rejection. Therefore, my suggestion for the author to improve this work is to reproduce it on publicly available data / model / code and release them for future version.

**Justification For Why Not Lower Score:**

N/A

---

### Decision · Program_Chairs · 2024-01-16

Reject